# Toward a Coronavirus Knowledge Graph

**DOI:** 10.3390/genes12070998

**Published:** 2021-06-29

**Authors:** Peng Zhang, Yi Bu, Peng Jiang, Xiaowen Shi, Bing Lun, Chongyan Chen, Arida Ferti Syafiandini, Ying Ding, Min Song

**Affiliations:** 1Department of Computer Science and Technology, Tsinghua University, Beijing 100084, China; zhangp31@tsinghua.edu.cn; 2Department of Information Management, Peking University, Beijing 100871, China; buyi@pku.edu.cn; 3Beijing Knowledge Atlas Technology Co., Ltd., Beijing 100043, China; peng.jiang@aminer.cn (P.J.); shixiaowen_win@163.com (X.S.); bing.lun@aminer.cn (B.L.); 4School of Information, University of Texas at Austin, Austin, TX 78701, USA; chongyanchen_hci@utexas.edu (C.C.); ying.ding@ischool.utexas.edu (Y.D.); 5Department of Library and Information Science, Yonsei University, Seoul 03722, Korea; afsyafiandini@yonsei.ac.kr; 6Dell Medical School, University of Texas at Austin, Austin, TX 78712, USA

**Keywords:** knowledge management applications, knowledge base management, knowledge engineering methodologies

## Abstract

This study builds a coronavirus knowledge graph (KG) by merging two information sources. The first source is Analytical Graph (AG), which integrates more than 20 different public datasets related to drug discovery. The second source is CORD-19, a collection of published scientific articles related to COVID-19. We combined both chemo genomic entities in AG with entities extracted from CORD-19 to expand knowledge in the COVID-19 domain. Before populating KG with those entities, we perform entity disambiguation on CORD-19 collections using Wikidata. Our newly built KG contains at least 21,700 genes, 2500 diseases, 94,000 phenotypes, and other biological entities (e.g., compound, species, and cell lines). We define 27 relationship types and use them to label each edge in our KG. This research presents two cases to evaluate the KG’s usability: analyzing a subgraph (ego-centered network) from the angiotensin-converting enzyme (ACE) and revealing paths between biological entities (hydroxychloroquine and IL-6 receptor; chloroquine and STAT1). The ego-centered network captured information related to COVID-19. We also found significant COVID-19-related information in top-ranked paths with a depth of three based on our path evaluation.

## 1. Introduction

The COVID-19 pandemic has caused nearly 1.28 million deaths worldwide (as of 6 December 2020) [1]. The disease has affected many human sectors worldwide and prompted scientists to explore the topic more extensively. Consequently, the number of scientific publications related to COVID-19 has increased sharply since 2020. Several bibliometric studies focused on the COVID-19 literature and aimed to understand the knowledge flow and trends [2]. However, there has not been much research exploring in-depth knowledge unit analysis (e.g., biological entities-level explorations), especially relationships between knowledge units. This limitation might prevent us from detecting the full knowledge flow in studies. Therefore, we require solid knowledge representation with clear definitions of knowledge unit relationships for greater understanding.

This study proposes a framework to merge two independent datasets Analytical Graph (AG) and CORD-19 [3] (with significant knowledge overlaps) into a new, larger knowledge graph (KG). We build the KG to promote more profound knowledge retrieval and in-depth knowledge mining. AG is a subgraph generated from multiple biomedical KGs, while CORD-19 contains coronavirus-related scientific publications. We adopt PubTator [4], a popular biological entity extraction tool, to extract entities from the CORD-19 literature. However, we cannot explicitly capture the relationships among extracted knowledge units. Therefore, we use AG to enrich the relationships among entities, yielding a more comprehensive, global-wise knowledge base, covering coronavirus entities and their “contexts”.

The new KG contains at least 21,700 genes, 2500 diseases, 94,000 phenotypes, and other biological entities (e.g., compound, species, and cell lines). We use 27 types of relationships in our KG and natural language processing techniques, such as entity recognition, semantic disambiguation, and knowledge merge. This KG could be used widely in the future. It functions as an essential knowledge base for related scientific research and development, while benefiting from knowledge retrieval and in-depth knowledge mining. This new KG acquires and integrates coronavirus-related information into an ontology and enables researchers to apply reasoning to derive new knowledge according to defined rules.

We evaluated the KG’s usability for information extraction using our pathfinding framework, which retrieves several paths with different depths. It calculates the path score based on the similarity distance between two nodes in every relationship found in the path. First, we transform nodes into vector values using a word vector transformation model. Then, we calculate the similarity distance between nodes using cosine similarity. We use a pre-trained word2vec model built using PubMed^®^ and PubMed Central^®^ (PMC) texts [5].

## 2. Related Work

Since the coronavirus outbreak in 2020, many researchers have focused on building a coronavirus KG. Domingo-Fernández et al. [6] built a KG of COVID-19 pathophysiology. Their KG is sourced from 145 related research articles, yielding approximately 4000 nodes, 9400 relationships, and 10 entity types (e.g., proteins, genes, chemicals). The researchers claimed this KG identified more than 300 candidate drugs currently proposed or investigated for COVID-19. Nevertheless, their KG scale seems small (only extracted from 145 research articles), and the graph does not cover sufficient information for COVID-19 research. Lu Wang et al. [3] extracted knowledge from the existing coronavirus literature to improve knowledge discovery. Their KG (http://blender.cs.illinois.edu/covid19/, accessed on 6 December 2020), through August 2020, has been updated daily. Ge et al. [7] proposed a novel, data-driven drug repositioning framework that enables the discovery of a potential therapeutic agent to treat COVID-19. This framework is crucial to building a KG containing interactions, associations, and similarities of drug human targets and virus targets. However, this KG only has 6200 drugs, 2500 human targets, and 404 virus targets.

Richardson et al. [8] established an ego-centric KG for baricitinib to analyze whether it is a potential drug for COVID-19. Xu et al. [9] proposed the PubMed KG, which extracts bio-entities from 29 million PubMed abstracts and integrates them with Authority, Semantic Scholar, and four additional resources. The previous efforts in other related studies [10,11] present preliminary explorations of COVID-19 KGs. Nevertheless, most of these studies include a small number of biological entities in their KGs that limit future studies.

When building a KG, researchers often rely on multiple sources instead of just one. For example, Sun et al. [12] extracted information from the text and added media files such as images into a KG. Before adding the image file into the KG, they transformed the image file into a vector value. Data fusion steps include data transformation, duplicate detection, and data integration. Each step requires natural language processing techniques, such as entity extraction, recognition, and relationship definition.

In graph theory, the shortest path problem aims to identify path problems between two nodes in a graph using the minimum sum of weights. We can obtain the possible paths between the two entities in the KG using the shortest path problem. Those paths symbolize two nodes’ relatability, knowledge fundamental for further research. The shortest path problem relies on edge weight initialization in the graph. Edge weight can represent either the amount of effort required to travel or the capacity to be transported. 

Because the KG structure depicts the relationship between entities, we assume the edge weight as the cost or similarity between entities. Previous studies calculated entity or term similarity using the Jaccard index [13] and term co-occurrence [14]. Despite promising results, they did not perform well in measuring term similarity for several conditions. Other researchers [15] proposed a similarity measurement by transforming words into a vector dimension. They refer to this transformation process as word embedding. One study on word embedding in topic segmentation [16] concluded that, depending on the choice of model, Word2Vec [17] could produce a more accurate vector representation than LSA and GloVe.

Our research differs from the existing literature because we produce a larger and more updated coronavirus KG. Furthermore, we demonstrate that the nodes in our retrieved ego network are highly correlated with COVID-19 and use case studies to confirm the possibility of using our proposed KG for further knowledge discovery.

## 3. Datasets

We incorporate an existing KG and bibliographic dataset—AG (Section 3.1) and CORD-19 (Section 3.2)—and merge them into one KG. In this section, we introduce the details of the two datasets.

### 3.1. Analytical Graph (AG)

AG is a subgraph generated from multiple chemogenomics repositories. In AG, compound data are obtained from ChEMBL [18], PubChem [19], and UniChem [20]. Protein and gene data are obtained from Ensembl [21], UniProt [22], TCRD [23] (Target Central Resource Database), ExplorEnz [24], and Gene Ontology [25]. Gene and disease data are obtained from DisGeNet [26]. Disease and phenotype data are obtained from UMLS [27]. Organism data are obtained from Disbiome [28]. Tissue data are obtained from neXtProt [29]. Pathway data are obtained from Reactome [30]. Side effects and adverse effect data are obtained from SIDER [31], STRING [32], Offsides [33], and STITCH [34].

This study adopts the latest version of AG. Table 1 presents the descriptive statistics of nodes (biological entities) and edges (relationships) in AG. Compounds have the most entities in AG, with 588,820 nodes, followed by phenotypes with 96,924 and genes with 19,946. Table 2 presents the statistical descriptions of 27 relationship types from AG. Given the high number of nodes for compounds and genes, we expect many relationships between those two.

### 3.2. CORD-19

In March 2020, the Allen Institute of AI and other leading research groups released a COVID-19 Open Research dataset, covering coronavirus-related scientific publication bibliographic metadata (COVID-19 Open Research Dataset [3]). According to its version released on 3 April 2020, there are approximately 47,000 publications in 1951–2020 from different sources, including (1) PubMed’s PMC open access corpus, (2) COVID-19 research articles from a corpus maintained by the WHO, and (3) bioRxiv and medRxiv preprints. Therefore, we adopt the following query terms for (1) and (3): “COVID-19” OR Coronavirus OR “Coronavirus” OR “2019-nCoV” OR “SARS-CoV” OR “MERS-CoV” OR “Severe Acute Respiratory Syndrome” OR “Middle East Respiratory Syndrome”.

For these publications, we extract biological entities mentioned in their titles and abstracts using PubTator [4], a web-based text mining tool for pre-annotating biological entities [35]. We obtain the recognized biological entities in each scientific publication and their types (e.g., gene, chemical, species, and mutation) with this toolkit. Table 3 presents the descriptive statistics of each biological entity. We identify 16,487 diseases, 8677 chemicals, 7080 genus, 5596 species, 703 protein mutations, and other types of biological entities. For the CORD-19 dataset, we extract entity co-occurrence relationships based upon the PubTator [4] extraction results. If two entities co-occur in the paper title or abstract, their number of occurrences increases by one. We extracted 260,295 co-occurrence relationships from the CORD-19 dataset.

## 4. Merging Different KGs

Before merging the two KGs, we must disambiguate biological entity names carefully. We used two external data sources to help with the disambiguation process: Wikidata and ref.txt. Wikidata is a large-scale, multilingual encyclopedia knowledge base that contains over 25 million entities (as of 2017) and their relationships [36]. Records in Wikidata link to Wikipedia data. Both robots and humans contribute to Wikidata collections. Furthermore, ref.txt is a reference file provided in the CORD-19 dataset. The CORD-19 dataset provides entities with their synonyms and annotations. 

First, we extracted entities and their types from both AG and CORD-19. Then, we grouped entities based on their types. Next, we mapped each entity to terms mentioned in Wikidata and ref.txt based on string similarities. Since there might be uncertainty and variability issues in the merging process, we ensured that we only merged entities of similar types. In addition, we carried out several evaluations by manually checking similarity measurement results to obtain the best threshold value for string similarity score.

If two or more entities link to the same term, we merged those entities into one. We mapped 161,324 entities from AG and CORD-19 with terms from Wikidata and ref.txt. We present examples of disease and gene synonyms in Table 4 and Table 5. For example, if we found “pneumocystis infections Wegener’s granulomatosis” in AG or CORD-19 records, we flagged it as “pneumocystis Carinii infection”. Similarly, for gene-type entities, if we found “msg1” in AG or CORD-19 records, we flagged it as “cited1”.

After disambiguating medical entities in AG and CORD-19, we merged those KGs and built a new KG. Table 6 presents the descriptive statistics of the newly merged KG. Previously, there were 27,026 gene entities: 19,946 from AG (Table 1) and 7080 from CORD-19 (Table 3). However, in the newly merged KG, there are only 21,761, indicating duplicate entities that we merged. A similar case also occurred for phenotype entities: there were 96,924 nodes in AG, but only 94,251 in the newly merged graph.

## 5. Cases

Based on the built KG, we present two cases to demonstrate KG usage. The first case illustrates an ego-centered network, a subgraph of the established KG (https://covid-19.aminer.cn/kg/?lang=en, accessed on 1 May 2021). The second case presents the path details of two biological entities.

### 5.1. Ego-Centered Subgraph

We illustrate a subgraph from our established KG in Figure 1. Each node in the subgraph represents a biological entity. Node labels represent entity names and biological types (e.g., chemical, gene, and disease). We defined different colors for different biological (node) types. The size of the nodes is proportional to their degree (the number of connected nodes). Each edge represents a relationship between two entities (e.g., co-occurrence, gene and cellular components, and gene-to-gene relationship). The label indicates the relationship type.

We chose the angiotensin-converting enzyme (ACE) gene, encoded as ACE, which has 40% overall identity to ACE-2 and is positively related to COVID-19 [37]. ACE-2 counters the related ACE activity by reducing angiotensin-II and increasing angiotensin-(1–7), making it a promising drug target for treating cardiovascular diseases. ACE-2 activators are also potential COVID-19 treatments, according to their popularity in the COVID-19 literature [38]. As depicted in Figure 1, we discovered that other entities such as SARS2 and PaO2 are highly related to COVID-19 because PaO2 reflects arterial oxygen tension and COVID-19 damages the lung.

### 5.2. Path

This section evaluates each path by scoring it using the similarity distance between nodes and verifying the information given in the top-ranked paths. We calculated the similarity distance using cosine similarity on vector representation from the vector transformation model [5]. First, we retrieved the shortest paths from the source node to the target node with several depths from the KG. We used three depths: two, three, and four. Paths with a depth of two have three nodes: one source node, one target node, and one node in between. Paths with a depth of three have two nodes in between, and paths with a depth of four have three nodes.

Second, for each path, we calculated and summed the cosine similarity between nodes. Third, we measured the cosine similarity between nodes using their vector values obtained from [5]. Finally, we sorted paths (separately for each depth) based on the sum of cosine similarity values between nodes. Then, we analyze whether the information given in the top-ranked paths is accurate. The top-ranked paths are paths with the top 95 percentile score from the distribution. We illustrate the evaluation process in Figure 2.

In this example, we analyzed two paths: (1) between IL-6 receptor and hydroxychloroquine and (2) between STAT1 and chloroquine. IL-6 receptor and STAT1 are both related to immune systems and COVID-19. We found one path with a depth two, 202 paths with a depth of three, and 600 paths with a depth of four for IL-6 receptor and hydroxychloroquine. We also found 62 paths with a depth of two, 642 paths with a depth of three, and 435 paths with a depth of four between STAT1 and chloroquine. We found evidence of gene–gene, gene–disease, and compound–disease relationships from those paths. We analyzed the evidence found in top-ranked paths for each depth and compared them with DisGeNet [26] and DrugBank [39].

#### 5.2.1. IL-6 Receptor and Hydroxychloroquine

This section discusses the characteristics of paths between two entities: IL-6 receptor and hydroxychloroquine. IL-6 is Interleukin 6, an interleukin that functions as both a pro-inflammatory cytokine and an anti-inflammatory myokine. IL-6 inhibitors may ameliorate severe lung tissue damage caused by cytokine release in patients with severe COVID-19 infections. Hydroxychloroquine is a medication used to prevent and treat malaria in areas where malaria remains sensitive to chloroquine. Other usage includes the treatment of rheumatoid arthritis (RA), lupus, and porphyria cutanea tarda (PCT).

Common side effects of hydroxychloroquine consumption include vomiting, headache, changes in vision, and muscle weakness. Severe side effects may include allergic reactions, vision problems, and heart problems. Although we cannot exclude all risks, it remains a treatment for rheumatic disease during pregnancy. Companies sell hydroxychloroquine under the brand name Plaquenil (among others).

The list of top-ranked paths (top 95 percentile) with a depth of two, three, and four based on the shortest path algorithm of the graph to find and calculate the score of all paths from IL-6 receptor to hydroxychloroquine is presented in Table 7. A subgraph from top-ranked paths with a depth of three is illustrated in Figure 3. 

We concluded that the Ebola virus infection co-occurs with IL-6 receptor and hydroxychloroquine from the evidence found in paths with a depth of two. We assume the first relationship between IL-6 receptor and the Ebola virus is accurate because experiments in [40] concluded that the elevated concentration of IL-6 in plasma during the symptomatic phase is a non-fatal Ebola virus infection marker. Furthermore, we found a supported argument in [41] for the second relationship between hydroxychloroquine and the Ebola virus.

Based on the Drugbank dataset [39], we also found that combinations with hydroxychloroquine can decrease the Ebola Zaire vaccine’s therapeutic efficacy (live, attenuated). Even though each relationship is correct, we cannot identify a potential relationship between IL-6 receptor and hydroxychloroquine from paths with a depth of two. Moreover, there is no significant relationship between the Ebola virus and COVID-19, except that both are pandemic diseases.

There are ten paths in the top-ranked path category with a depth of three. We found twenty-one different relationships (node–relationship–node). In addition to the Ebola virus, we found another disease that appeared in top-ranked paths: autoimmune diseases. We found that the relatedness between COVID-19 and autoimmune disease is more substantial compared to the Ebola virus. A recent report found autoimmune diseases in COVID-19 patients [42]. We also found the RA disease in top-ranked paths. RA disease is related to autoimmune disease [26], and because RA patients are more likely to catch certain infections, they have a higher chance of getting COVID-19. DisGeNet [26] also reported that IL-6 receptor is a biomarker in RA. 

We found three more compounds in top-ranked paths with a depth of three: chloroquine, amodiaquine, and quinoline. Drugbank [39] reported that amodiaquine and chloroquine are currently in clinical trials for COVID-19. We also found several genes related to autoimmune disease or RA, such as cd4, ccr7, il17a, il10, and cd83. DisGeNet [26] reported that cd4 is a therapeutic factor for arthritis infection, but we could not find it in the top-ranked paths with a depth of three.

There are 29 paths in the top-ranked path with a depth of four and 54 different relationships. Based on top-ranked paths with a depth of four, we found two types of diseases: HIV infections and malaria. HIV infection is an autoimmune disease that may have a higher risk in COVID-19. For malaria, there is a probability of misdiagnosis in COVID-19 and malaria [43]. Compared to paths with a depth of three, paths with a depth of four involve more nodes but are less related to COVID-19 information.

#### 5.2.2. STAT1 and Chloroquine

STAT1 is the primary transcription factor activated by interferons (IFNs) vital to normal immune responses, particularly viral, mycobacterial, and fungal pathogens [44]. An innate immune response is a defense strategy that includes physical, chemical, and cellular level defenses. Type I IFNs are a critical component of this response. In COVID-19 cases caused by the SARS-CoV-2 N protein that inhibits the phosphorylation of STAT1 and STAT2, the conditions also suppress IFN signaling [45]. Chloroquine, also known as Chlorochin and Aralen [46], has been studied to treat and prevent COVID-19.

Chloroquine is an aminoquinoline primarily used to prevent and treat malaria in areas where it remains sensitive. Chloroquine is also vital as an anti-inflammatory agent in RA and lupus therapy. The list of top-ranked paths (top 95 percentile) with a depth of two, three, and four is presented in Table 8. A subgraph from top-ranked paths with a depth of three is illustrated in Figure 4.

In contrast to IL-6 receptor–hydroxychloroquine, there are three paths with a depth of two in the STAT1–chloroquine case. We found six different relationships in paths with a depth of two. However, it is challenging to obtain information from paths with a depth of two. We found a “chloroquine–co_occur–weight loss” relationship in paths with a depth of two, but we could not find supporting evidence. The information obtained from paths with a depth of two is more related to mice.

In top-ranked paths with a depth of three, we found 31 paths and 15 different relationships. We found eight different gene nodes between the head and tail nodes in top-ranked paths with a depth of three. The gene nodes are MX1, MX2, ISG15, OAS1, OAS2, JAK1, OASL, and EIF2AK2. According to [47], the MX1, ISG15, and OAS2 interferon-stimulated genes are potential candidates for drug targets in COVID-19 treatments. Furthermore, we found other evidence to support the relatedness of OAS1, JAK1, and OASL with COVID-19 [48,49,50]. However, we could not find supporting evidence for MX2 and EIF2AK2.

When we evaluated the top-ranked paths with a depth of four in the STAT1–chloroquine case, we found 51 paths and 56 different relationships. We found other genes in the top-ranked paths with a depth of four, such as USP18, A226V, SOCS, and LY96. USP18 is a differentially expressed gene (DGS) in COVID-19 cases [49]. However, we could not find supporting evidence for hyperglycemiaA226V, SOCS, and LY96. Aside from co-occurring genes, we also found 15 diseases: dengue hemorrhagic fever (DHF), John Cunningham (JC) virus infection, dengue shock syndrome, alphavirus infections, hypoxia, pneumococcal pneumonia, pleural effusion, myalgia, asthma, empyema, hyperglycemia, bronchiectasis, arthralgia, bronchiolitis, and pneumonia.

In COVID-19 cases, there is a higher incidence of bilateral pneumonia and pleural effusion [51]. The most common symptoms at diagnosis were coughs, myalgia, dyspnea, fever, and chills [52]. In some cases, acute bronchiolitis with mucous membrane exfoliation, accumulation of bronchiolar secretions, and bronchiolar epithelial metaplasia occurred [53]. A Spanish COVID-19 case series in Barcelona found that myalgia or arthralgia is a protective factor against ICU admission and death [54]. Moreover, underlying lung disease, especially asthma, has recently been associated with a higher risk of hospitalization [55].

As with the IL-6 receptor–hydroxychloroquine case, we can find more information using a higher depth (depth four) but obtain fewer significant paths than using depth three. However, results from STAT1–chloroquine are slightly different as chloroquine is also related to many other diseases. Therefore, in the STAT1-chloroquine case, there are more irrelevant nodes and information to COVID-19 extracted from top-ranked paths.

## 6. Conclusions

This study built a coronavirus KG by merging two existing datasets: AG and CORD-19. The combination of the two datasets enriches the KG with more entities. However, further analysis is needed to illustrate that those entities contribute to understanding the COVID-19 disease context. We analyzed our built KG using an ego network analysis for nodes, such as ACE, SARS, and PaO2. From the retrieved ego network, we can discover the high relatedness between those nodes and COVID-19.

We attempted pathfinding using a defined head and tail node to confirm KG usability for further knowledge discovery. We found that we could obtain paths with significant relationships using word-embedding and distance similarity between nodes. We also found that using a depth of three in both IL-6 receptor–hydroxychloroquine and STAT1–chloroquine cases resulted in more information related to COVID-19.

In the future, we plan to update this KG with more recent coronavirus publications. We also plan to include more related knowledge resources to enrich the graph. We will perform a further experiment in the COVID-19 domain query search for knowledge discovery using the built KG. We will explore more paths on scoring methods and missing link prediction.

## Figures and Tables

**Figure 1 genes-12-00998-f001:**
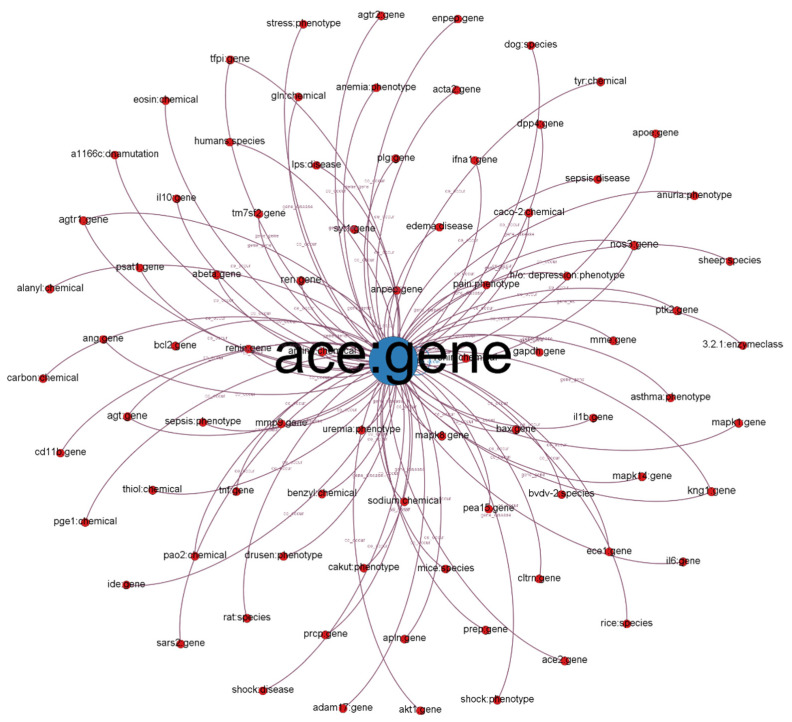
Subgraph from established KG with ACE: gene as the center node.

**Figure 2 genes-12-00998-f002:**
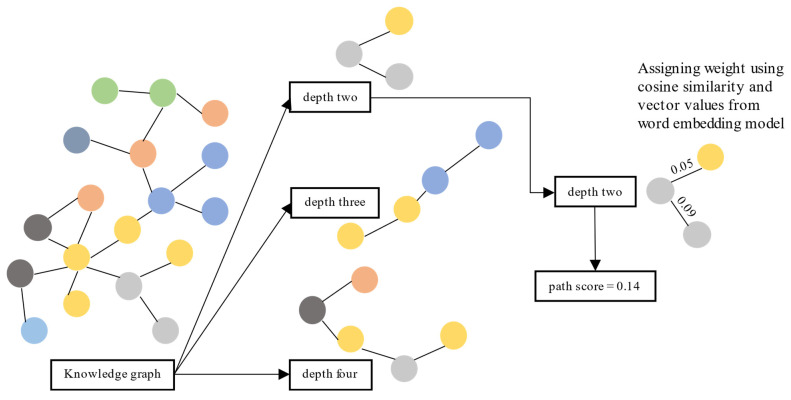
Evaluation process for KG usability using a path ranking framework.

**Figure 3 genes-12-00998-f003:**
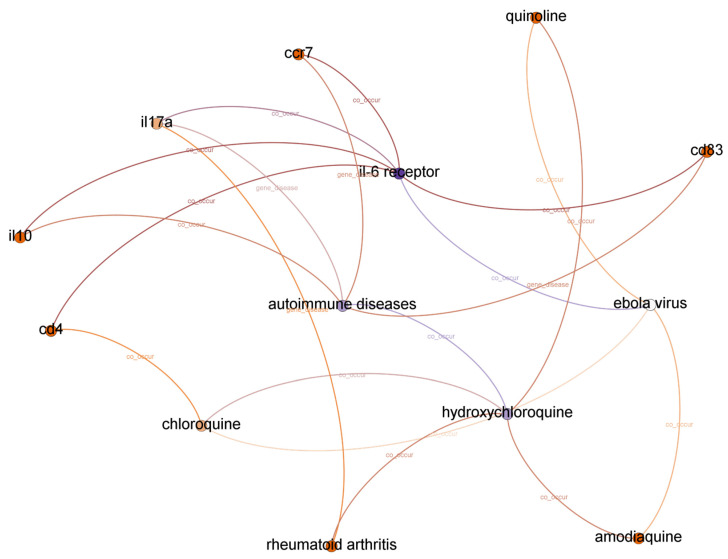
Subgraph from top-ranked paths with a depth of three in IL-6 receptor and hydroxychloroquine case.

**Figure 4 genes-12-00998-f004:**
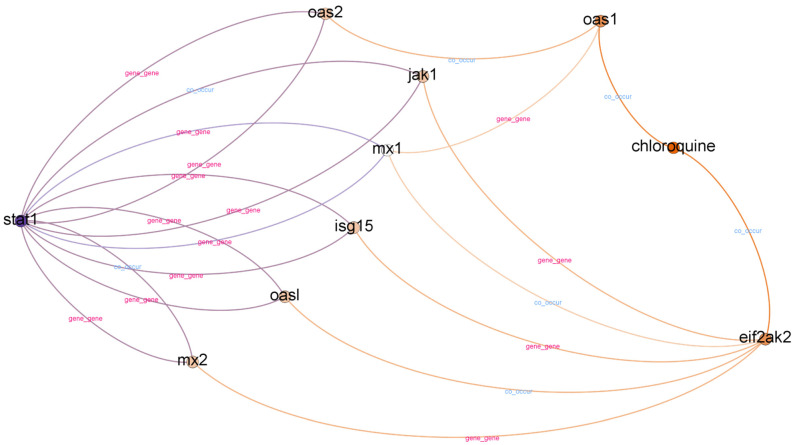
Subgraph from top-ranked paths with a depth of three in STAT1 and chloroquine case.

**Table 1 genes-12-00998-t001:** Descriptive statistics of entities extracted from AG.

Type	Number of Entities
Compound	588,820
Phenotype	96,924
Gene	19,946
Biological process	12,313
Enzyme class	8077
Gene Ontology (GO)	6002
Pathway	2205
Organism	1419
Tissue	94

**Table 2 genes-12-00998-t002:** Descriptive statistics of relationships extracted from AG.

Type	Number of Relationships
COMPOUND_GENE	1,331,963
GENE_DISEASE	648,348
COMPOUND_ADVERSE_EFFECT	453,684
GENE_GENE	381,389
IS_A_PHENOTYPE	247,563
GENE_BIOLOGICALPROCESS	177,898
GENE_CELLULARCOMPONENT	117,323
GENE_MOLECULARFUNCTION	92,316
GENE_TISSUE	48,900
PATHWAY_GENE	40,632
COMPOUND_INDICATION	33,868
INSTANCE_OF	21,936
IS_A_EC	8069
CHANGES_WITH	6952
REPURPOSED_INDICATION	6609
PATHWAY_COMPOUND	5869
PATHWAY_CELLULARCOMPONENT	4608
PART_OF	3705
GENE_EC	2331
PATHWAY_CONTAINS_PATHWAY	2245
CANONICAL_TARGET	2080
POSITIVELY_REGULATES	1439
NEGATIVELY_REGULATES	1278
REGULATES	1199
HAS_PART	338
OCCURS_IN	111
GENE_GO	1

**Table 3 genes-12-00998-t003:** Descriptive statistics of entities extracted from CORD-19.

Type	Number of Entities
Disease	16,487
Chemical	8677
Gene	7080
Species	5596
Protein mutation	703
Single nucleotide polymorphisms (SNPs)	162
DNA mutation	155
Cell line	68
Genus	15
Strain	2

**Table 4 genes-12-00998-t004:** Examples of disease synonyms.

Disease Name	Synonym
pneumocystis Carinii infection	pneumocystis infections Wegener’s granulomatosis
acute lymphoblastic leukemia	acute lymphocytic leukemia
adult respiratory distress syndrome	respiratory distress syndrome, adult malignant gliomas
bunyavirus infection	Bunyaviridae infections
breast cancer	breast carcinoma
thyroid cancer	thyroid neoplasm

**Table 5 genes-12-00998-t005:** Examples of gene synonyms.

Gene Name	Synonym
msg1	cited1
pla2s	pla2g2a
amyloid precursor protein	app caveolin 1
bcl-w	bcl2l2
ro52	trim21
timp	timp1
dead box helicase 5	ddx5
aconitase 2	aco2

**Table 6 genes-12-00998-t006:** Descriptive statistics of the merged KG.

Type	Number of Entities
Compound	588,820
Phenotype	94,251
Gene	21,761
Biological process	12,120
Enzyme class	8077
GO	5737
Chemical	4817
Species	3060
Disease	2565
Pathway	2201
Organism	1419
Protein mutation	678
SNP	162
DNA mutation	148
Tissue	94
Cell line	39
Genus	15
Strain	2

**Table 7 genes-12-00998-t007:** Top-ranked paths from IL-6 receptor to hydroxychloroquine.

Depth	Path	Score
2	Il-6_receptor--co_occur--ebola virus--co_occur--hydroxychloroquine	0.271000
3	il-6 receptor--co_occur--ebola virus--co_occur--chloroquine--co_occur--hydroxychloroquine	0.983559
3	il-6 receptor--co_occur--ebola virus--co_occur--quinoline--co_occur--hydroxychloroquine	0.967936
3	il-6 receptor--co_occur--ebola virus--co_occur--amodiaquine--co_occur--hydroxychloroquine	0.964461
3	il-6 receptor--co_occur--cd4--co_occur--chloroquine--co_occur--hydroxychloroquine	0.902787
3	il-6 receptor--co_occur--il17a--gene_disease--rheumatoid arthritis--co_occur--hydroxychloroquine	0.895744
3	il-6 receptor--co_occur--il17a--gene_disease--autoimmune diseases--co_occur--hydroxychloroquine	0.887945
3	il-6 receptor--co_occur--il10--gene_disease--autoimmune diseases--co_occur--hydroxychloroquine	0.884826
3	il-6 receptor--co_occur--il10--co_occur--autoimmune diseases--co_occur--hydroxychloroquine	0.884826
3	il-6 receptor--co_occur--cd83--gene_disease--autoimmune diseases--co_occur--hydroxychloroquine	0.882492
3	il-6 receptor--co_occur--ccr7--gene_disease--autoimmune diseases--co_occur--hydroxychloroquine	0.860170
4	il-6 receptor--co_occur--cd83--gene_gene--cd86--gene_disease--hiv infections--co_occur--hydroxychloroquine	1.275014
4	il-6 receptor--co_occur--ccr2--gene_gene--ccr1--gene_disease--malaria--co_occur--hydroxychloroquine	1.262865
4	il-6 receptor--co_occur--ccr2--gene_gene--ccr3--gene_disease--hiv infections--co_occur--hydroxychloroquine	1.247009
4	il-6 receptor--co_occur--ccr2--co_occur--cx3cr1--gene_disease--hiv infections--co_occur--hydroxychloroquine	1.229658
4	il-6 receptor--co_occur--ccr2--gene_gene--cxcr3--gene_disease--hiv infections--co_occur--hydroxychloroquine	1.225782
4	il-6 receptor--co_occur--ccr2--gene_gene--ccr5--gene_disease--hiv infections--co_occur--hydroxychloroquine	1.214904
4	il-6 receptor--co_occur--ccr2--gene_gene--cxcr5--gene_disease--hiv infections--co_occur--hydroxychloroquine	1.198998
4	il-6 receptor--co_occur--ccr2--gene_gene--ccr6--gene_disease--hiv infections--co_occur--hydroxychloroquine	1.185817
4	il-6 receptor--co_occur--gsto1--gene_gene--prdx2--gene_disease--malaria--co_occur--hydroxychloroquine	1.177270
4	il-6 receptor--co_occur--ccr2--gene_gene--ccr3--gene_disease--malaria--co_occur--hydroxychloroquine	1.169309
4	il-6 receptor--co_occur--ccr2--gene_gene--cxcr3--gene_disease--malaria--co_occur--hydroxychloroquine	1.151890
4	il-6 receptor--co_occur--ccr2--gene_gene--ccl7--gene_disease--hiv infections--co_occur--hydroxychloroquine	1.149486
4	il-6 receptor--co_occur--ccr2--gene_gene--cxcr1--gene_disease--hiv infections--co_occur--hydroxychloroquine	1.149216
4	il-6 receptor--co_occur--ccr2--gene_gene--cxcl10--gene_disease--malaria--co_occur--hydroxychloroquine	1.144627
4	il-6 receptor--co_occur--ccr2--gene_gene--ccl22--gene_disease--hiv infections--co_occur--hydroxychloroquine	1.138721
4	il-6 receptor--co_occur--ccr2--gene_gene--cxcr2--gene_disease--hiv infections--co_occur--hydroxychloroquine	1.136466
4	il-6 receptor--co_occur--ccr2--gene_gene--ccr7--gene_disease--malaria--co_occur--hydroxychloroquine	1.126956
4	il-6 receptor--co_occur--ccr2--gene_gene--cxcl8--gene_disease--malaria--co_occur--hydroxychloroquine	1.123747
4	il-6 receptor--co_occur--ccr2--gene_gene--ccl20--gene_disease--hiv infections--co_occur--hydroxychloroquine	1.119462
4	il-6 receptor--co_occur--ccr2--co_occur--ccl2--gene_disease--hiv infections--co_occur--hydroxychloroquine	1.119136
4	il-6 receptor--co_occur--ccr2--gene_gene--ccl2--gene_disease--hiv infections--co_occur--hydroxychloroquine	1.119136
4	il-6 receptor--co_occur--gsto1--gene_gene--gstk1--gene_disease--malaria--co_occur--hydroxychloroquine	1.119127
4	il-6 receptor--co_occur--ccr2--gene_gene--ccl22--gene_disease--malaria--co_occur--hydroxychloroquine	1.117001
4	il-6 receptor--co_occur--ccr2--gene_gene--ccl2--gene_disease--malaria--co_occur--hydroxychloroquine	1.116415
4	il-6 receptor--co_occur--ccr2--co_occur--ccl2--gene_disease--malaria--co_occur--hydroxychloroquine	1.116415
4	il-6 receptor--co_occur--ccr2--gene_gene--cx3cl1--gene_disease--hiv infections--co_occur--hydroxychloroquine	1.092402
4	il-6 receptor--co_occur--ccr2--gene_gene--cxcl12--gene_disease--hiv infections--co_occur--hydroxychloroquine	1.092225
4	il-6 receptor--co_occur--ccr2--gene_gene--cxcl10--gene_disease--hiv infections--co_occur--hydroxychloroquine	1.086009
4	il-6 receptor--co_occur--ccr2--gene_gene--cxcr6--gene_disease--hiv infections--co_occur--hydroxychloroquine	1.081086

**Table 8 genes-12-00998-t008:** Top-ranked paths from STAT1 to chloroquine.

Depth	Path	Score
2	stat1--co_occur--weight loss--co_occur--chloroquine	0.700930
2	stat1--co_occur--mice--co_occur--chloroquine	0.593730
2	stat1--co_occur--mnv--co_occur--chloroquine	0.559861
3	stat1--gene_gene--oasl--co_occur--eif2ak2--co_occur--chloroquine	1.402100
3	stat1--gene_gene--oasl--co_occur--eif2ak2--co_occur--chloroquine	1.402100
3	stat1--gene_gene--oas2--co_occur--oas1--co_occur--chloroquine	1.347831
3	stat1--gene_gene--oas2--co_occur--oas1--co_occur--chloroquine	1.347831
3	stat1--gene_gene--mx2--co_occur--eif2ak2--co_occur--chloroquine	1.345132
3	stat1--co_occur--mx2--co_occur--eif2ak2--co_occur--chloroquine	1.345132
3	stat1--gene_gene--mx2--co_occur--eif2ak2--co_occur--chloroquine	1.345132
3	stat1--gene_gene--mx2--gene_gene--eif2ak2--co_occur--chloroquine	1.345132
3	stat1--co_occur--mx2--gene_gene--eif2ak2--co_occur--chloroquine	1.345132
3	stat1--gene_gene--mx2--gene_gene--eif2ak2--co_occur--chloroquine	1.345132
3	stat1--co_occur--isg15--gene_gene--eif2ak2--co_occur--chloroquine	1.267028
3	stat1--gene_gene--isg15--gene_gene--eif2ak2--co_occur--chloroquine	1.267028
3	stat1--gene_gene--isg15--gene_gene--eif2ak2--co_occur--chloroquine	1.267028
3	stat1--co_occur--mx1--gene_gene--oas1--co_occur--chloroquine	1.248431
3	stat1--gene_gene--mx1--gene_gene--oas1--co_occur--chloroquine	1.248431
3	stat1--co_occur--mx1--co_occur--oas1--co_occur--chloroquine	1.248431
3	stat1--gene_gene--mx1--co_occur--oas1--co_occur--chloroquine	1.248431
3	stat1--gene_gene--mx1--co_occur--oas1--co_occur--chloroquine	1.248431
3	stat1--gene_gene--mx1--gene_gene--oas1--co_occur--chloroquine	1.248431
3	stat1--co_occur--jak1--co_occur--eif2ak2--co_occur--chloroquine	1.242361
3	stat1--gene_gene--jak1--gene_gene--eif2ak2--co_occur--chloroquine	1.242361
3	stat1--gene_gene--jak1--co_occur--eif2ak2--co_occur--chloroquine	1.242361
3	stat1--co_occur--jak1--gene_gene--eif2ak2--co_occur--chloroquine	1.242361
3	stat1--gene_gene--jak1--co_occur--eif2ak2--co_occur--chloroquine	1.242361
3	stat1--gene_gene--jak1--gene_gene--eif2ak2--co_occur--chloroquine	1.242361
3	stat1--gene_gene--mx1--gene_gene--eif2ak2--co_occur--chloroquine	1.233766
3	stat1--co_occur--mx1--co_occur--eif2ak2--co_occur--chloroquine	1.233766
3	stat1--gene_gene--mx1--co_occur--eif2ak2--co_occur--chloroquine	1.233766
3	stat1--gene_gene--mx1--gene_gene--eif2ak2--co_occur--chloroquine	1.233766
3	stat1--co_occur--mx1--gene_gene--eif2ak2--co_occur--chloroquine	1.233766
3	stat1--gene_gene--mx1--co_occur--eif2ak2--co_occur--chloroquine	1.233766
3	stat1--gene_disease--jc virus infection--co_occur--myalgia--co_occur--arthralgia--co_occur--chloroquine	1.402100
4	stat1--gene_disease--jc virus infection--co_occur--dengue shock syndrome--co_occur--arthralgia--co_occur--chloroquine	1.015049
4	stat1--gene_disease--jc virus infection--co_occur--oas2--co_occur--oas1--co_occur--chloroquine	0.973442
4	stat1--gene_disease--jc virus infection--co_occur--hyperglycemia--co_occur--metformin--co_occur--chloroquine	0.966955
4	stat1--gene_disease--jc virus infection--co_occur--mx2--gene_gene--eif2ak2--co_occur--chloroquine	0.906810
4	stat1--gene_disease--jc virus infection--co_occur--mx2--co_occur--eif2ak2--co_occur--chloroquine	0.878100
4	stat1--gene_disease--jc virus infection--co_occur--phenazopyridine--co_occur--monensin sodium--co_occur--chloroquine	0.878100
4	stat1--gene_disease--jc virus infection--co_occur--alphavirus infections--co_occur--arthralgia--co_occur--chloroquine	0.876126
4	stat1--gene_disease--jc virus infection--co_occur--a226v--co_occur--arthralgia--co_occur--chloroquine	0.793003
4	stat1--gene_disease--jc virus infection--co_occur--empyema--co_occur--pneumonia--co_occur--chloroquine	0.787724
4	stat1--gene_disease--jc virus infection--co_occur--pleural effusion--co_occur--pneumonia--co_occur--chloroquine	0.777438
4	stat1--gene_disease--jc virus infection--co_occur--pneumococcal pneumonia--co_occur--pneumonia--co_occur--chloroquine	0.739655
4	stat1--gene_disease--jc virus infection--co_occur--ly96--gene_gene--eif2ak2--co_occur--chloroquine	0.736294
4	stat1--gene_disease--jc virus infection--co_occur--usp18--gene_gene--eif2ak2--co_occur--chloroquine	0.710234
4	stat1--gene_disease--jc virus infection--co_occur--hypoxia--co_occur--arthralgia--co_occur--chloroquine	0.708574
4	stat1--gene_disease--jc virus infection--co_occur--isg15--gene_gene--eif2ak2--co_occur--chloroquine	0.707835
4	stat1--gene_disease--jc virus infection--co_occur--bronchiectasis--co_occur--bronchiolitis--co_occur--chloroquine	0.702962
4	stat1--gene_disease--jc virus infection--co_occur--asthma--co_occur--bronchiolitis--co_occur--chloroquine	0.697532
4	stat1--gene_disease--jc virus infection--co_occur--dhf--co_occur--arthralgia--co_occur--chloroquine	0.694675
4	stat1--gene_disease--jc virus infection--co_occur--socs1--co_occur--eif2ak2--co_occur--chloroquine	0.692914

## Data Availability

Publicly available datasets were analyzed in this study. These data can be found here: https://lfs.aminer.cn/misc/cokg19/CORD19-AG.csv.zip (accessed on 30 December 2020).

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
