# Peer review of "Toward a Coronavirus Knowledge Graph"

_genes, 2021, doi:10.3390/genes12070998_

Round 1

Reviewer 1 Report

1) In this article authors build a coronavirus knowledge graph (KG). Following are some of my concerns with this study.

2) Authors see that the the number of genes in the new graph is 21,761, while the numbers in AG and CORD-19 are 19,946 and 7,080, respectively. It will be extremely useful for readers to have a list of list discrepancies as a supplemental file.

3) Are the Wikidata and the ref.txt gene identifiers exact match? Please confirm.

4) Can authors map gene lists in Descriptive statistics of AG extracted relationships either to Wikidata or ref.txt?

5) Can authors provide more details on score calculation in table 7?

6) How can thresholds be applied for above scores in ranking? Any reference for it?

7) It would be best to have a online web tool with aggregated knowledge graph and if authors can provide this link in the manuscript?

Thanks,

Author Response

In this article authors build a coronavirus knowledge graph (KG). Following are some of my concerns with this study.

1. Authors see that the the number of genes in the new graph is 21,761, while the numbers in AG and CORD-19 are 19,946 and 7,080, respectively. It will be extremely useful for readers to have a list of list discrepancies as a supplemental file.

Thank you for your nice inputs. We provide the data for this research in this link https://lfs.aminer.cn/misc/cokg19/CORD19-AG.csv.zip. We also provide the supplementary information on this page https://covid-19.aminer.cn/kg/?lang=en. We mentioned this information in the manuscript (page 7, line 184 and page 17, line 361).

2. Are the Wikidata and the ref.txt gene identifiers exact match? Please confirm.

Wikidata and ref.txt have several similar terms (medical entities). We matched entities that we found in AG and CORD-19 using a string matching method with a specific threshold value that we found after several experiments. As a result, we managed to map most of our medical entities with ones in Wikidata and ref.txt. However, there are some entities that we could not map to those data.

3. Can authors map gene lists in Descriptive statistics of AG extracted relationships either to Wikidata or ref.txt?

The relationships came from the annotator framework. We use Wikidata and ref.txt to disambiguate nodes (entities) that we found in AG and CORD-19 datasets. The complete data description is available in https://covid-19.aminer.cn/kg/?lang=en. We already referred to that page in the manuscript.

4. Can authors provide more details on score calculation in table 7?

We revised and explained how we got values in Table 7, page 9, line 209 to 219.

“…First, we retrieved the shortest paths from the source node to the target node with several depths from the KG. We used three depths, depth two, three, and four. Paths with depth two have three nodes, one source node, one target node, and one node in between. Paths with depth three have two nodes in between, and paths with depth four have three nodes. Second, for each path, we calculated the cosine similarity between nodes and summed it. Third, we measured the cosine similarity between nodes using their vector values obtained from [4]. Lastly, we sorted paths (we did it separately for each depth) based on the sum of cosine similarity values between nodes. Then, we analyze whether the information given in the top-ranked paths is accurate or not. The top-ranked paths are paths with the top 95 percentile score from the distribution….”

5. How can thresholds be applied for above scores in ranking? Any reference for it?

We did not use any threshold in the path ranking. Instead, we only analyzed paths in the top 95 percentile in this study.

6. It would be best to have a online web tool with aggregated knowledge graph and if authors can provide this link in the manuscript?

For more data visualization and utility, you can access this page https://covid-19.aminer.cn/kg/?lang=en.

Reviewer 2 Report

 I have little concerns about the grammar and vocabulary of the manuscript; therefore, the improvement of the language is highly needed. Overall, there is a significant novelty and clear implications of the particular results that make this manuscript suitable for the publication of this rank.  I hope the author will re-write or improve the scientific language and grammar in this manuscript along with the figure quality improvement.

Author Response

Thank you for your inputs; we fixed several grammar errors and rewrote some sentences. We do hope the newly edited manuscript will meet the publication standard. We also improved the figure qualities in the manuscript.

Reviewer 3 Report

The authors use two existing knowledge graphs related to COVID-19 and merge them to a new graph. In two examples of subgraphs they demonstrate the usability of the new graph.

Major issues: the process of merging the two existing graphs is not sufficiently described. The authors should present more details on software or own code they used for merging. In addition, the merging result is based on some necessary preprocessing steps (e.g. matching of synonymous words as listed in Table 4 and 5) which drags along uncertainty and varibility to the resulting merged graph. Can the authors describe, how subgraphs of the two examples and their interpretation change, when merging parameters are changed.

Minor issues:
1) Table description are very short. Could be more detailed.
2) Some references are not correctly linked in the text. E.g., page 4, line 150; page 7, line 202.

Author Response

The authors use two existing knowledge graphs related to COVID-19 and merge them to a new graph. In two examples of subgraphs they demonstrate the usability of the new graph.

Major issues: the process of merging the two existing graphs is not sufficiently described. The authors should present more details on software or own code they used for merging. In addition, the merging result is based on some necessary preprocessing steps (e.g. matching of synonymous words as listed in Table 4 and 5) which drags along uncertainty and varibility to the resulting merged graph. Can the authors describe, how subgraphs of the two examples and their interpretation change, when merging parameters are changed.

Thank you for your comments; we have added new paragraphs on page 6, lines 163 – 167, explaining the merging process.

“…First, we extracted entities and entity types from both AG and CORD-19. Then, we grouped entities based on their entity types. Next, we mapped each entity to terms (with similar type) mentioned in Wikidata and ref.txt based on string similarities. If two or more entities link into the same term, we merged those entities as one. We managed to map 161,324 entities from AG and CORD-19 with terms from Wikidata and ref.txt…”

We agreed that there might be some uncertainty and variability in the merging process. However, in the merging process, we made sure that we only merged entities of similar types. Moreover, we also run the string similarity step several times to get the best threshold value and most precise results. Therefore, we are quite certain that our approach is effective in eliminating those aspects.

Minor issues

  • Table description are very short. Could be more detailed. Thank you, we added more explanations for Table 1-8.
  • Some references are not correctly linked in the text. E.g., page 4, line 150; page 7, line 202. Thank you for the correction, we already checked those links and corrected them.

Round 2

Reviewer 3 Report

In principle, the answer on the uncertainty issue is acceptable ("We agreed that there might be some uncertainty and variability in the merging process. However, in the merging process, we made sure that we only merged entities of similar types. Moreover, we also run the string similarity step several times to get the best threshold value and most precise results. "), but this information should also be provided to the readers as part of the method and result section, in particular the process of finding the threshold.

Author Response

# Reviewer 3

In principle, the answer on the uncertainty issue is acceptable ("We agreed that there might be some uncertainty and variability in the merging process. However, in the merging process, we made sure that we only merged entities of similar types. Moreover, we also run the string similarity step several times to get the best threshold value and most precise results. "), but this information should also be provided to the readers as part of the method and result section, in particular the process of finding the threshold.

Thank you for the suggestion. We have added a short explanation in the manuscript (page 6, line 169-173).

… Since there might be uncertainty and variability issues in the merging process, we ensured that we only merged entities of similar types. In addition, we carried out several evaluations by manually checking similarity measurement results to obtain the best threshold value for string similarity score …